# The Mechanisms of Polysaccharides from Tonic Chinese Herbal Medicine on the Enhancement Immune Function: A Review

**DOI:** 10.3390/molecules28217355

**Published:** 2023-10-31

**Authors:** Zhiyi Xie, Ninghua Jiang, Minqiu Lin, Xinglishang He, Bo Li, Yingjie Dong, Suhong Chen, Guiyuan Lv

**Affiliations:** 1College of Pharmaceutical Science, Zhejiang Chinese Medical University, Hangzhou 310053, China; 2Collaborative Innovation Center of Yangtze River Delta Region Green Pharmaceuticals, Zhejiang University of Technology, Huzhou 313200, China; 3Zhejiang Provincial Key Laboratory of TCM for Innovative R & D and Digital Intelligent Manufacturing of TCM Great Health Products, Huzhou 313200, China; 4The Second Affiliated Hospital of Jiaxing University, Jiaxing 314000, China; jiangninghua@zjxu.edu.cn

**Keywords:** traditional Chinese medicine, polysaccharides, autoimmunity, mechanisms of action

## Abstract

Tonic Chinese herbal medicine is a type of traditional Chinese medicine, and its primary function is to restore the body’s lost nutrients, improve activity levels, increase disease resistance, and alleviate physical exhaustion. The body’s immunity can be strengthened by its polysaccharide components, which also have a potent immune-system-protecting effect. Several studies have demonstrated that tonic Chinese herbal medicine polysaccharides can improve the body’s immune response to tumor cells, viruses, bacteria, and other harmful substances. However, the regulatory mechanisms by which various polysaccharides used in tonic Chinese herbal medicine enhance immune function vary. This study examines the regulatory effects of different tonic Chinese herbal medicine polysaccharides on immune organs, immune cells, and immune-related cytokines. It explores the immune response mechanism to understand the similarities and differences in the effects of tonic Chinese herbal medicine polysaccharides on immune function and to lay the foundation for the future development of tonic Chinese herbal medicine polysaccharide products.

## 1. Introduction

Immunity is a physiological process that helps the body eliminate pathogens (bacteria, viruses, etc.) and tumor cells to keep the body healthy. The emergence of novel coronavirus in recent years has posed a significant threat to the immune function of the human body. Rapid spread, fast variation, and complex sequelae were the characteristics of the novel coronavirus. The novel coronavirus currently comes in various variants, and each variant requires a particular vaccine to fight against it. The mutation speed of the virus is frequently faster than that of the vaccine, and the method of resisting the novel coronavirus only by injecting the vaccine has little effect because the research and development of a novel coronavirus variant vaccine require several experiments and time investment. Intriguingly, several research studies have revealed a strong relationship between the recovery of novel coronaviruses and immune function [1,2]. Therefore, it is crucial to enhance and regulate autoimmunity to resist the intrusion of toxic and harmful substances such as viruses.

Tonic Chinese herbal medicine polysaccharides are active polysaccharides extracted from traditional Chinese medicine and have the benefits of green environmental protection and reduced adverse effects. Polysaccharides used in traditional Chinese medicine have reportedly been found to have a range of physiological activities, including antitumor [3], antiviral [4], antibacterial [5], and immunomodulatory [6,7] activities. However, polysaccharides as immunopotentiators in traditional Chinese medicine are a popular area of research. Numerous pharmacological studies have demonstrated that most of the polysaccharides used in traditional Chinese medicine—which have been shown to improve immune function—come from tonic Chinese herbal medicine. This particular form of traditional Chinese medicine works to restore the body’s lost nutrients, improve activity levels, increase disease resistance, and alleviate physical exhaustion. Tonic Chinese herbal medicine can be divided into four categories based on their specific functions. They are tonic Chinese herbal medicine for tonifying vital energy, the backbone of the entire system, such as *Atractylodes macrocephala* Koidz., *Astragalus membranaceus*, *Acanthopanax senticosus*, and *Ziziphus jujuba*; tonic Chinese herbal medicine for tonifying blood, such as *Angelica sinensis*; tonic Chinese herbal medicine for tonifying body function and energy, such as *Cordyceps sinensis* and *Epimedium brevicornu*; and tonic Chinese herbal medicine for tonifying body stores substances, including essence, blood, and body fluid, such as *Dendrobium officinale*, *Polygonatum sibiricum*, and *Lycium chinense*. Figure 1 shows the categories of tonic Chinese herbal medicine. These polysaccharides from traditional Chinese medicine are exceptional in regulating the human immune system, but the specific mechanism by which they do so has not been thoroughly explored.

In this review, we counted the number of tonic Chinese herbal medicine polysaccharides that have been shown to improve immune function over the last decade in PubMed and Web of Science library. We also discuss the impact of polysaccharides found in tonic Chinese herbal medicine on the immune system from four dimensions: immune organs, immune cells, immune-related cytokines, and immune regulatory mechanisms. We can further investigate the similarities and differences of tonic Chinese herbal medicine polysaccharides on the regulation of immune function and enhance the application in medical treatment and healthy food by understanding the effects of tonic Chinese herbal medicine polysaccharides on the immune system.

## 2. The Effects of Tonic Chinese Herbal Medicine Polysaccharides on Organs of the Immune System

The immune organ, which also serves as the site of immune cell formation, significantly influences how well the body’s immune system works. The two categories of immune organs include central immune organs (thymus, bone marrow, or bursa of Fabricius) and peripheral immune organs (spleen and lymph nodes). The immune system dramatically benefits from immune organs with a normal structure and development.

### 2.1. The Effects of Tonic Chinese Herbal Medicine Polysaccharides on the Thymus, Spleen, and Bursa of Fabricius

The bursa of Fabricius in birds and the thymus are the sites of lymphocyte differentiation and maturation. The spleen, the largest peripheral immune organ and where the immune response takes place, contains many T and B lymphocytes. Immune organ growth and immune function are reflected in the organ index, which is a crucial index. The immune function is better when the immune organ index is higher. [8]. A large number of studies have reported that tonic Chinese herbal medicine polysaccharides can increase the organ index of immune organs. After LBP (a crude polysaccharide from *Lycium chinense*) treatment, the thymus index and spleen index of normal mice increased significantly by 22.19% and 44.05% [9], respectively. APS (a polysaccharide from *Astragalus membranaceus*) [10,11] and ASPS (a crude polysaccharide from *Acanthopanax senticosus*) [12] can increase the index of the spleen and thymus in cancer model animals. LBP [13], PPS (a purified polysaccharide from *Cordyceps sinensis*) [14], PSP (a crude polysaccharide from *Polygonatum sibiricum*), and PSP3 (a purified polysaccharide from PSP) [15,16,17] could increase the thymus index and spleen index of cyclophosphamide-immunosuppressed mice. PSP [18], APS [19], RAMPS60c, and RAMPStp (purified polysaccharides from RAMPS, a polysaccharide from *Atractylodes macrocephala*) [20] could increase the organ index of the thymus, bursa of Fabricius, and spleen of chickens. This shows that traditional Chinese medicine polysaccharides can enhance the immune function of the body by increasing the organ index.

### 2.2. The Effects of Tonic Chinese Herbal Medicine Polysaccharides on the Mucosal Immune System

Mucosal immunity is the immunity on the mucosal surface of the cavity tracts (respiratory tract, digestive tract, genitourinary tract, etc.) through which the animal communicates with the outside environment, directly or indirectly, and serves as a crucial link for the body to resist infection. The mucosal immune system is the first line of defense against the invasion of pathogenic microorganisms. The intestinal mucosal immune system depends on Pyle’s collecting lymph nodes (PP nodes). The size and number of PP nodules can indicate the state of intestinal mucosal immunity. DOP-W3-b (a purified polysaccharide from Dendrobium nobile) can increase the number of PP nodes in the intestine and promote the secretion of cytokines in mesenteric lymph nodes (MLNs). It can effectively regulate the immune activity of the intestinal mucosa by changing the structure of the intestinal mucosa [21].

## 3. The Effects of Tonic Chinese Herbal Medicine Polysaccharides on Immune Cells

Immune response or immune response-related cells include immune cells, macrophages, dendritic cells, T and B lymphocytes, and natural killer (NK) cells. The most crucial professional antigen-presenting cells are macrophages and dendritic cells, which may successfully present antigens to subsequent T lymphocytes and activate the subsequent immune response [22]. T lymphocytes mediate cellular immunity by secreting cytokines, which help the body resist the invasion of toxic and harmful substances. B lymphocytes play a crucial role in mediating humoral immunity by producing specific antibodies that attach to the antigen and phagocytose via macrophages. NK cells can bind and eliminate target cells.

### 3.1. The Effects of Tonic Chinese Herbal Medicine Polysaccharides on Macrophages

Innate immune cells, known as macrophages, are found throughout the body in various organs derived from monocytes. Macrophages are nonspecifically recognized, treated, and presented to T cells to stimulate T-cell proliferation and differentiation and trigger an additional immune response after being stimulated by foreign bodies with exogenous antigens. The immune function of macrophages can be shown in their capacity for phagocytosis and morphological changes. In vitro studies showed that *Dendrobium* CPs (a crude polysaccharide from *Dendrobium*) [23], ADPS-1a, ADPS-3a (purified polysaccharides from *Angelica sinensis*) [24], PSPC (a purified polysaccharide from PSP), and PSPW (a wine-processed PSP) [25] could increase the phagocytic activity of macrophages. In vivo studies have found that APS (an alcohol-soluble polysaccharide extracted from *Astragalus membranaceus*) [26], DSP (a purified polysaccharide from *Dendrobium nobile*) [27], and PSP [15] can enhance the immune function of macrophages.

Macrophages can be divided into two types: M1 macrophages, which are induced by LPS or IFN-γ, secrete cytokines such as TNF-α, IL-12, and IL-1β to promote inflammation, and their markers are the expression of CD80, CD86, and MHC-II on the cell surface; and M2 macrophages, which are induced by IL-4, secrete IL-10 in vivo to exert an anti-inflammatory effect, and their marker is the expression of CD206 on the cell surface [28,29]. CMPB90-1 (a purified polysaccharide from *Cordyceps sinensis*) [30], ISAg (a purified polysaccharide from *Angelica sinensis*) [31], RAMPtp (a purified polysaccharide from *Atractylodes macrocephala* Koidz.) [32], and PG2 (a purified polysaccharide from APS) [33] can upregulate the expression of CD86 and MHC-II in macrophages and promote the polarization of macrophages into type M1. Further study showed that the expression and phagocytosis of CD80, CD86, and MHC-II in macrophages were significantly enhanced by the inclusion of APS with a Poly (lactic-co-glycolic acid) (PLGA) carrier [34]. In addition, another study found that CMPB90-1 can reset the M2 phenotype of tumors to the M1 phenotype of tumor killing [30].

### 3.2. The Effects of Tonic Chinese Herbal Medicine Polysaccharides on Dendritic Cells

The only professional antigen-presenting cells that can in vivo activate initial T cells (naive T cells) are dendritic cells (DCs). The treated antigens are exposed to the extracellular surface by DC cells after attaching them to the major histocompatibility complex. At this point, immature DCs transform into mature DCs, which can activate T lymphocytes and trigger a specific type of immunity. Therefore, DCs are the link between innate immunity and specific immunity [35]. CTAB-modified PSP-Cubs/OVA [36] and ISAg [31] could significantly activate dendritic cells. ASP-PLGA-PEI (polyethylenimine-coated PLGA nanoparticles containing ASP, a polysaccharide from *Angelica sinensis*, system) [37], APS [38,39], and LBP [40] can upregulate the expression of MHC I/II, CD40, CD80, and CD86 on DCs and stimulate their maturation, and APS [41] can also induce morphological changes in human blood monocyte-derived dendritic cells (MDDCs). Another study found that intranasal administration of APS [42] increased the number of DCs in mesenteric lymph nodes.

### 3.3. The Effect of Tonic Chinese Herbal Medicine Polysaccharide on T Lymphocytes

T lymphocytes can be divided into three groups based on their different functions: (1) helper T lymphocytes, (2) cytotoxic T lymphocytes, and (3) regulatory/suppressor T cells. The cell surface marker CD4+ identifies helper T cells (Th cells). Antigens and cytokines can stimulate Th cells to differentiate into Th1 cells, Th2 cells, and other cell subsets [43]. To eliminate tumor cells, activated CD4+ T cells can identify tumor cells with specificity. These cells can then activate and recruit effector cells, such as CD8+ T cells. Cytotoxic T lymphocytes, also known as CD8+ T lymphocytes, activate CD8+ T cells to function as effector cells and induce apoptosis of tumor cells by releasing cytokines and exerting cytotoxicity [44,45]. Regulatory T lymphocytes, also called suppressor T lymphocytes, are cells with highly expressed CD25 on the cell surface and can inhibit cellular and humoral immunity [46]. Only after activation do these lymphocytes carry out immune functions. Studies have found that tonic Chinese herbal medicine polysaccharides can activate lymphocyte proliferation and promote lymphocytes from the G0 phase to the G2-S-M phase. RAMPtp, RAMPStp, and RAMPS60c [20,47,48] can promote lymphocytes to enter the S phase and G2/M phase and enhance the specific immunity of the body. The effect of tonic Chinese herbal medicine polysaccharides on T lymphocytes is shown in Figure 2.

The number of mature T lymphocytes in the body is usually determined by measuring CD3+. JP (a crude polysaccharide from *Ziziphus jujuba*) [49], APS4 (a purified polysaccharide from APS) [10], and GXG (a purified polysaccharide from DOP) [50] could increase the number of CD3+ cells on the surface and the total number of mature T lymphocytes in vivo. Polysaccharides of tonic Chinese herbal medicine can regulate the number and distribution of CD4+ T cells and CD8+ T cells to enhance immunity. LBP and LBPL (LBP liposomes) can help antigens activate CD4+ and CD8+ T cells at the same time [40,51]. APS4 can effectively regulate the proportion of CD3+, CD4+, and CD8+ T cells in the thymus, peripheral blood, and spleen of S180 tumor-bearing mice in a dose-dependent manner [10]. APS [19], RAMPS60c, and RAMPStp [47] can increase the number of CD4+ and CD8+ T lymphocytes in ND chickens. PSP [15] can improve the abnormal location, distribution, and quantity of CD4+ and CD8+ T lymphocytes in the spleen induced by cyclophosphamide. In addition, some traditional Chinese medicine polysaccharides can activate CD4+ T lymphocytes but not promote CD8+ T lymphocytes. CSP (a crude polysaccharide from *Cordyceps sinensis*) [52] could increase the number of CD4+ cells but decrease the number of CD8+ cells in cyclophosphamide-immunosuppressed mice.

After stimulation by antigens and cytokines, Th cells mainly differentiate into Th1, Th2, and Th17 cells. Th1 cells specifically express T-bet transcription factors, Th2 cells specifically express GATA-3 transcription factors, and Th17 cells specifically express ROR-γt transcription factors [53,54]. CSP [55] can promote the expression of transcription factors (T-bet, GATA-3, and RoR-γt) in Th cells and increase the number of Th1, Th2, and Th17 cells. GXG [50] stimulated the transformation of Th cells into Th1 and Th2 cells. JPC (JP conjugates) [56] and ASP-PLGA-PEI [57] had a stronger stimulating bias for Th cells to differentiate into Th1 cells but did not promote the differentiation of Th2 cells. CSP [52] can specifically promote the expression of RoR-γt, and APS [58] can increase the proportion of Th17 cells and Th17/Treg cells.

### 3.4. The Effect of Tonic Chinese Herbal Medicine Polysaccharide on B Lymphocytes

B lymphocytes are the primary cells that mediate humoral immunity in the body, and they primarily rely on antigens to induce the release of antibodies to exert their immune function [59]. Antigen-induced stimulation of B lymphocytes results in their transformation into plasma cells capable of producing antibodies [10]. The hemolysin value, which can represent the humoral immunity of the body, can be used to show the ability of B-cell proliferation and differentiation. It has been reported that GXG [50] and RAMP [60] can increase the number of B lymphocytes, thereby enhancing humoral immunity.

### 3.5. The Effects of Polysaccharides from Tonic Chinese Herbal Medicine on NK Cells

NK cells can react directly with cancer cells without any assistance [26]. ASP can inhibit the growth of H22 tumor cells, 4T1 tumor cells, and B16F10 mouse melanoma cells by increasing the activity of NK cells [11,26,42]. It is suggested that ASP can inhibit the growth of cancer cells by activating NK cells. ISAg [31], ADP (a crude polysaccharide from *Angelica dahurica*) [61], JCP [56], PSP, and PSP3 [15,16] can activate NK and enhance the immune function of the body.

## 4. The Effect of Tonic Chinese Herbal Medicine Polysaccharides on Cytokines

Cytokines are small molecular peptides or glycoproteins that are primarily synthesized and secreted by immune cells. They can mediate cell interaction and perform many biological functions, including regulating immune responses and participating in inflammatory responses. Immunoglobulin (IgG, IgA, IgM, etc.), interleukin (IL-1, IL-2, IL-4, etc.), tumor necrosis factor (TNF-α), transforming growth factor (TGF-β), nitric oxide-related cytokines, and so on are the key cytokines that play a role in the immune system.

### 4.1. The Effect of Tonic Chinese Herbal Medicine Polysaccharides on Immunoglobulin

B lymphocytes primarily perform their immune function by secreting antibodies, a class of immunoglobulin that can bind specific antigens, such as IgG, IgA, IgM, and IgE. The body produces IgG, the most prevalent antibody in serum, as a result of a response. It has a high affinity and is widely distributed in the body. In serum, IgA is the second most abundant antibody after IgG, and it is crucial for antibacterial and antiviral immunological activities. IgM is an antibody produced by the body’s initial response that is crucial in early-stage resistance to foreign harmful substances. IgE is an essential antibody in type I hypersensitivity and is mainly produced by Th2 cells and their secreted cytokines [62].

APS [63] can increase the level of three serum antibodies (IgG, IgA, and IgM) at the same time. ASPS [64] significantly increased the levels of IgA and IgM in serum but had no significant effect on IgG. PEI-MM-PLGA-DP/OVA [65] and PSP [36] all promoted the production of OVA-specific IgG antibody in the serum.

In the body, IgG has different subsets, such as IgG1, IgG2a, IgG2b, and IgG3. Among them, IgG2a, IgG2b and IgG3 are mainly induced by cytokines produced by Th1 cells. IgG1 is mainly induced by cytokines produced by Th2 cells. LBPL-OVA [51], APS, and APSL [66] enhance cellular immunity by stimulating mice to produce specific antibodies IgG, IgG1, and IgG2a. LMw-APS (a purified polysaccharide from APS) [67] can increase the content of specific antibodies IgG, IgG1, and IgG2b in the serum of mice immunized with rP-HSP90C. APS [38] could promote the production of IgG1 in tumor-bearing mice but did not promote the production of IgG2a.

SIgA, an immunoglobulin secreted by plasma cells (IgA cells) in the lamina propria of the intestinal mucosa, is the main effector of the mucosal adaptive immune system. It can prevent the invasion of harmful pathogens, neutralize toxins, enzymes, and viruses in the intestine, and enhance the immune function of the intestine [68,69]. It has been reported that LBP [9], GXG [50], and DOP-W3-b [21] can increase the level of sIgA in the intestine. RAMPS [70], APS [71], and JP [49] can increase the content of IgA plasma cells and sIgA antibody in the intestinal tract and play a role in enhancing intestinal mucosal immunity.

### 4.2. The Effect of Tonic Chinese Herbal Medicine Polysaccharides on Interleukin

Leukocytes, crucial immune system components, are the primary source of IL. Th1 cells primarily secrete IL-2, which facilitates the cellular immune response. Th2 cells predominantly secrete IL-4, IL-5, IL-6, IL-10, and IL-13, which can help B-cell activation and mediate humoral immunity [53]. Th17 cells primarily secrete IL-17, IL-21, and IL-22, which are crucial in developing autoimmunity and inflammation. Macrophages primarily secrete cytokines such as IL-1 (including IL-1α and IL-1β), IL-6, IL-8, and IL-12, which are crucial for resisting bacterial and viral invasion, sterilization, and scavenging cell damage [72,73].

RAMPtp can not only increase the expression of IL-1α, IL-1β, IL-2, IL-3, IL-4, IL-6, IL-10, IL-12, and IL-13 in mouse spleen cells in vitro [47], but it also significantly promotes the secretion of IL-1α and IL-21 in bovine lymphocytes [74]. In vivo, it has been found that RAMP can increase the levels of IL-2 and IL-6 in the serum of cyclophosphamide-immunosuppressed mice [75]. Another study found that RAMPS can increase the level of IL-6 in the serum of mice immunized with the FMDV O vaccine [70].

In vitro cell experiments showed that APS could promote the expression of IL-6 in RAW264.7 cells [76]. In vivo, LMw-APS can upregulate the levels of IL-2, IL-4, IL-10, and IL-12 in the serum of HSP90C-injected mice [67]. APS could not only increase the levels of IL-2, IL-4, and IL-6 in the serum of Newcastle immunized chickens [19], but it also promoted IL-2 and IL-4 in the serum of HBV-immunized mice [39]. Furthermore, APS can upregulate the levels of IL-2 in the serum of 4T1 tumor-bearing mice and H22 hepatoma mice [11,77] and promote the levels of IL-4 and IL-10 in the serum of tumor-bearing mice treated with a focused ultrasound [38]. In addition to the mouse model, APS not only increases the level of IL-17 in broilers infected with necrotizing enteritis (NE) [58], but it also promotes the expression of IL-1β, IL-8, and IL-10 in the spleen, kidney, liver. and intestinal tissues of SVCV-infected crucian carp [78].

A number of animal studies have shown that PSP, PSPC, and PSPW can promote the levels of IL-2, IL-6, and IL-8 in the serum of cyclophosphamide-injected mice [15,17,25]. However, another study found that PSP and PSP3 could inhibit the contents of IL-4 and IL-10 in the serum of mice immunosuppressed by cyclophosphamide [16]. Wang et al. [79] established a mouse model of blood deficiency syndrome by acetyl phenyl hydrazine (APH) and cyclophosphamide (CTX). SPSP (a polysaccharide from steam-processed *Polygonatum sibiricum*) treatment can increase the level of IL-6 in serum.

A large number of in vitro studies have reported that *Dendrobium* CPs [23], DOP [80], PEI-MM-PLGA-DP/OVA [65], DDP (a polysaccharide from *Dendrobium devonianum*) [81], UDP-1, FDP-1, FLP-1 (UDP-1: purified polysaccharide from unfermented *Dendrobium*; FDP-1: polysaccharide from fermented *Dendrobium*; FLP-1: polysaccharide from fermented FDP-1 liquid) [82], DOPA, DOPA-1, and DOPA-2 (purified polysaccharides from DOPA) [83] can induce RAW264.7 cells to produce IL-1, IL-1α, IL-1β, IL-4, and IL-6. Another cell culture experiment in vitro found that DOP-1-1 (a purified polysaccharide from DOP) could promote the production of IL-1β cytokines in THP-1 cells [84]. In vivo studies on Dendrobium polysaccharides showed that DSP [27] could increase the level of IL-6 in the serum of cyclophosphamide-immunosuppressed mice. GXG [50] and DOP-W3-b [21] can promote the levels of many kinds of interleukins in normal mice and increase the contents of IL-1α, IL-2, IL-4, IL-10, IL-12, IL-13, and IL-17 in serum, thus enhancing the immune function of the body.

In vitro cell culture experiments showed that ASP-PLGA-PEI could promote DCs to secrete IL-12p70 [37]. ASP-PLGA-PEI [57] and ADPs-1a and ADPs-3a [24] can promote the production of IL-1β, IL-12 and IL-6 cytokines in mouse peritoneal macrophages. In vivo, it was found that ADP could increase the content of IL-2 in the serum of H22 tumor-bearing mice [61].

In vivo, LBP could increase the levels of IL-1β, IL-2, and IL-6 in the serum of mice treated with cyclophosphamide [13], and promote the levels of IL-2 and IL-6 in the serum of normal mice [85].

CCP (a purified polysaccharide from CSP) can promote the secretion of IL-6 by macrophages [86]. CSP [52,55] and PPS [14] can increase the concentrations of IL-12, IL-4, IL-13, IL-6, IL-17, IL-10, IL-2, and IL-21 in the serum of cyclophosphamide-immunosuppressed mice.

In vivo, JP could increase the levels of IL-2, IL-4, and IL-10 in the serum of mice treated with cyclophosphamide [49], increase the level of IL-2 in the serum of chronic fatigue (CFS) rats, and decrease the level of IL-10 [56].

ASPS could increase the levels of IL-2 and IL-12 in the serum of S180, H22, and U14 tumor-bearing mice [12].

### 4.3. The Effects of Tonic Chinese Herbal Medicine Polysaccharides on Nitric Oxide-Related Cytokines

Nitric oxide synthase (iNOS) secretes nitric oxide (NO) when macrophages are stimulated by antigens. NO secreted by activated macrophages can produce cytotoxicity to bacteria, fungi, and tumor cells [87]. RAMPtp [32] and APS [76] can increase iNOS expression in mouse macrophages and increase the secretion of NO. In vitro, CAP, sCAP2 (purified polysaccharides from ADP) [88], ADP [61], RAMAP-1, RAMAP-2 and RAMAP-3 (purified polysaccharides from RAMP) [89], APS [90], DDP [81], UDP-1, FDP-1, FLP-1 [82], DOPA [83], PSPC, and PSPW [25] have been proven to promote NO secretion by macrophages and participate in the innate immune response.

Interferons (IFNs) are cytokines that have physiological activities, such as antiviral, antitumor activities, and immune regulation. IFN-γ, produced by activated T cells, NK cells, and other cells, is the most researched. IFN-γ has potent antitumor and antiangiogenic properties, and by regulating the expression of the *c-Myc* gene, it can slow the proliferation of cancer cells. Nitric oxide synthase, which can kill bacteria, tumor cells, and other harmful substances, is produced by macrophages when IFN-γ is present [91]. RAMPtp [47,74] and APS [42] have a remarkable ability to increase the expression of IFN-γ in lymphocytes in vitro. In vivo studies showed that PSP [18], PSPC, PSPW [25], LBP [13], JP [49], and DSP [27]. could increase the level of IFN-γ in the serum of mice injected with cyclophosphamide. APS and APSP [11,26,38,77] can promote the secretion of IFN-γ and kill tumor cells. APS can also increase the content of IFN-γ in the serum of mice immunized with OVA, the Newcastle disease vaccine, and the HBV vaccine [19,39,66]. It is suggested that APS can increase the level of IFN-γ and enhance immune function in many kinds of vaccine immunosuppressive animal models. Further study showed that the combination of APS and simvastatin could also significantly promote the secretion of IFN-γ in OVA-immunized mice [92].

### 4.4. The Effect of Tonic Chinese Herbal Medicine Polysaccharides on Transforming Growth Factor

TNF-β is mainly produced by lymphocytes, and its function is to promote cell differentiation but inhibit cell proliferation. APS [93] and LBP [94] can inhibit the expression of TGF-β to inhibit tumor growth. CSP [52,55] can increase the concentration of TGF-β 3 in the serum of mice immunized with cyclophosphamide. In addition, LBP [9,94] can increase the level of TGF-β in the serum of normal mice. RAMPS [70] can increase the level of TGF-β in the serum of mice immunized with the FMDV O vaccine. APS [78] can increase the level of TGF-β in the tissues of SVCV-infected crucian carp.

### 4.5. The Effect of Tonic Chinese Herbal Medicine Polysaccharides on Tumor Necrosis Factor

The cytokine TNF can kill tumor cells. Macrophages and T lymphocytes majorly produce them. TNF-α is the most prominent member of the TNF family, is crucial to the immune system, and performs an essential antitumor role [95]. A large number of studies have found that CCP [86], CAP, sCAP2 [88], *Dendrobium* CPs [23], RAMPtp [32], and PSP [96] can induce the production of TNF-α in macrophages. In vivo, ASPS [26], ADP [61], ISAg [31], and ASP [11,38] could increase the level of TNF-α in the serum of tumor-bearing mice. It is suggested that tonic Chinese herbal medicine polysaccharides can increase the level of TNF-α in tumor-bearing mice, thus killing tumor cells and improving immune function. Tonic Chinese herbal medicine polysaccharides can also improve the immunity of animal models with low immunity. LBP [13], JP [49], DSP [27], PSPC, PSPW [25], and CSP [52,55] could increase the level of TNF-α in serum and enhance the immune function of mice with myelosuppression induced by cyclophosphamide. SPSP [79] treatment can increase the level of TNF-α in a mouse model of blood deficiency syndrome.

### 4.6. The Effects of Tonic Chinese Herbal Medicine Polysaccharides on Other Related Cytokines

Macrophages generate lysozyme, which has antibacterial, anti-inflammatory, and antiviral properties. Lysozyme also works in conjugation with bacterial lipopolysaccharides to promote and enhance macrophage phagocytosis, reduce the role of endotoxin, and enhance the body’s resistance. Since lysozymes are an essential component of macrophages, acid phosphatase is a marker enzyme for them. Therefore, the level of acid phosphatase may be a good indicator of the degree of activation of macrophages [97]. PSPC and PSPW [25] can both increase acid phosphatase activity. APS and GLP (a polysaccharide from *Ganoderma lucidum*) could upregulate the lysozyme activity of pearl gentian grouper, and the combined effect was better [98].

The effect of tonic Chinese herbal medicine polysaccharides on the immune system is shown in Table 1. 

## 5. Study on the Mechanism of Tonic Chinese Herbal Medicine Polysaccharides to Enhancing Immunity

The mechanism of tonic Chinese herbal medicine polysaccharides in enhancing immunity is mainly related to MAPK, NF-κB, TLR, JAK-STAT, and other signaling pathways. The main signaling pathways affected by polysaccharides are shown in Figure 3.

### 5.1. Polysaccharides of Tonic Chinese Herbal Medicine Activate the MAPK Signaling Pathway to Enhance Immunity

Extracellular signal-regulated kinase (ERK), stress-activated protein kinase (JNK), and p38 are the three regulatory proteins that make up the category of serine/threonine protein kinases known as mitogen-activated protein kinases (MAPKs). According to reports, these three proteins function as the upstream pathways for cytokines such as TNF-α, IL-6, and IL-1β. The ERK signaling pathway, the p38 MAPK signaling pathway, and the JNK signaling pathway are the three different types of MAPK signaling pathways, according to the active regulatory proteins. The activation of the MARK signaling pathway significantly influences cellular immunity [99,100]. After macrophages were treated with RAMPtp [32], APS [76], ADPs-1a, and ADPs-3a [24], the phosphorylation levels of ERK, JNK, and p38 increased. It is suggested that the polysaccharides of traditional Chinese medicine can activate macrophages through the MAPK signaling pathway. RAMPtp [47,74] can enhance the transcriptional activity of AP-1, a signaling factor downstream of the JNK protein, and activate the JNK signaling pathway in lymphocytes. CCP [86] and CMPB90-1 [30] can activate the p38 MARK signaling pathway in macrophages and lymphocytes, respectively. DPFs (a purified polysaccharide from DOP) [101] induce apoptosis of HeLa cells through the p38MAPK signaling pathway. DOP-1-1 [84] can activate THP-1 cells through the ERK1/2 signaling pathway.

### 5.2. Tonic Chinese Herbal Medicine Polysaccharides Activates the NF-κB Signaling Pathway to Enhance Immunity

p65 is the activator of NF-κB. Phosphorylated p65 can be transferred from the cytoplasm to the nucleus, activating the NF-κB signaling pathway. Studies have shown that RAMPtp [32,47] can significantly promote the expression of cytoplasmic free NF-κB and the entry of NF-κB into the nucleus, suggesting that RAMPtp can activate the NF-κB signaling pathway. APS [76] can activate the expression level of p-p65 in RAW264.7 cells. DOP-1-1 [84] stimulates THP-1 cells through the NF-κB signaling pathway. The mechanism by which LBPL [40] activates DCs is through activating the NF-κB signaling pathway. CSP [54] and PPS [14] can promote the expression of p-IκB-α and NF-κB p65 in cyclophosphamide-immunosuppressed mice. The combined use of APS and sulfated EPS (a polysaccharide from *Epimedium*) can significantly increase the expression of NF-κB in the small intestine of piglets, and the effect is more significant than that of a single polysaccharide [102].

### 5.3. Polysaccharides of Tonic Chinese Herbal Medicine Activate the TLR Signaling Pathway to Enhance Immunity

The cell membrane or endosomal membrane contains Toll-like receptors (TLRs). The TLR family may identify pathogen-related molecules carried by various microorganisms, including viruses, bacteria, parasites, and fungi, triggering an immune response [103]. The immune mechanism of traditional Chinese medicine polysaccharides is most extensively researched through the TLR4 signaling pathway. Activating the TLR4 signaling pathway is crucial for immune function since it is the pathway upstream of NF-κB and MAPK. The TLR4 signaling pathway can be activated in two ways: MyD88-dependent and MyD88-independent [101,104,105]. TRAF6 is essential for the downstream IRAK4 and IRAK1 pathways that depend on MyD88 [106]. CSP [55], PPS [14], and CCP [86] can upregulate the expression of TLR4, MYD88 and TRAF-6 and enhance the activation of the MyD88-dependent TLR4/TRAF6 signaling pathway. LBP [40] can also activate DCs through the MyD88-dependent TLR4 signaling pathway. In addition, Bi et al. [30] found that CMPB90-1 can be used as a ligand of the TLR2 receptor to promote the release of calcium ions, thus activating the downstream pathway.

### 5.4. Polysaccharides of Tonic Chinese Herbal Medicine Activate the JAK-STAT Signaling Pathway to Enhance Immunity

JAKs can be activated by many cytokines, and downstream target genes can be activated by signal transducers and transcriptional activators (STATs), thus playing a regulatory role in cellular biological function [107]. It has been reported that RAMPtp [32,74] can upregulate the expression of p-JAK2, p-STAT1, and p-STAT3, indicating that it can activate the JAK-STAT signaling pathway in macrophages to improve immune ability. APS [37] can increase the phosphorylation of Jak2 and STAT3, suggesting that Angelica polysaccharides can promote DC maturation by activating the JAK2/STAT3 signaling pathway.

### 5.5. Tonic Chinese Herbal Medicine Polysaccharides Activate Other Signaling Pathways to Enhance Immunity

Akt kinase is an important downstream protein of PI3K that can regulate the development, survival, and function of immune cells. It has been found that the mechanism by which RAMPtp [32,74] activates macrophages and lymphocytes involves PI3K-Akt signaling pathways. In addition, another study found that CMPB90-1 [30] can activate the PI3K-Akt signaling pathway.

Ca^2+^ plays an important role in the proliferation and differentiation of lymphocytes. The increase in the intracellular Ca^2+^ level activates NFATs and initiates the transcription of specific cytokine genes [108]. Xu et al. [74] found that RAMPtp can increase the expression of p-NFAT4 in lymphocytes, suggesting that RAMPtp can activate the Ca^2+^ signaling pathway.

## 6. Safety Evaluation of Tonic Chinese Herbal Medicine Polysaccharides

Polysaccharides used in Chinese herbal tonics have a high level of safety, and no adverse reactions have yet been reported. Nevertheless, over-immunity might pose a risk, with cytokine storms being a frequent potential problem. Acute respiratory distress syndrome and multiple organ failure are often brought on by cytokine storms, which are the phenomena of rapid mass production of different cytokines such as TNF-α, IL-1, IL-6, IL-12, IFN-α, IFN-β, IFN-γ, MCP-1, and IL-8 in body fluids [109,110]. However, no research has discovered that the tonic Chinese herbal medicine polysaccharides can result in a cytokine storm, which may be connected to the advantages of Chinese medicine, including alleviating the effect, various targets, and overall regulation. The production of anti-inflammatory cytokines and M2 macrophage polarization by PG2 have been demonstrated to be effective treatments for cytokine storms caused by COVID-19 lung injury [111].

## 7. Discussion

Traditional Chinese medicine for medicinal use has a long history in China, but its constituents are complex and have various mechanisms of action. Tonic Chinese herbal medicine polysaccharides offer the advantages of stable composition, safe green, and non-toxic components as active components extracted from tonic Chinese herbal medicine. The effects and action sites of various polysaccharides vary, and several studies have demonstrated that tonic Chinese herbal medicine polysaccharides have unique advantages in enhancing immune function.

The tonic polysaccharides of *Polygonatum sibiricum*, *Astragalus membranaceus*, *Lycium chinense*, and other tonic Chinese herbal medicine polysaccharides can improve the organ index of the thymus, spleen, and other immune organs. In contrast, *Dendrobium officinale* polysaccharide can improve immune function by regulating the immune activity of the intestinal mucosa. The main impacts of polysaccharides on immune cells are as follows: (1) Important approaches for *Astragalus membranaceus* polysaccharides to improve immunity include promoting macrophage polarization, activating macrophages to secrete related cytokines, and enhancing macrophage phagocytic ability. (2) Improving the capacity of dendritic cells to process and present antigens and their number and maturation. (3) The development of cellular immune function dramatically depends on stimulating T lymphocyte maturation and differentiation and secreting cytokines such as IL-2, IFN-γ, and IL-4 to improve the subsequent immune response. (4) *Atractylodes macrocephala* and *Dendrobium officinale* polysaccharides can stimulate the maturation of B lymphocytes and the secretion of specific antibodies (IgG, IgA, and IgM) to exert humoral immune function. It can also increase the level of sIgG in the intestinal mucosa to safeguard the integrity of intestinal mucosa. (5) Increasing the capacity of NK cells for killing and producing cytotoxicity that affects the target cells directly. Polysaccharides might help improve immunity by increasing the expression of immunoglobulin (such as IgG, IgA, and IgM), interleukin (such as IL-1, IL-2, and IL-4), TNF-α, TGF-β, nitric oxide-related cytokines, and other molecules. Among these, an essential strategy for *Astragalus membranaceus* polysaccharide to improve immunity is increasing the expression of nitric oxide-related cytokines. Additionally, polysaccharides can help improve immunity by triggering the TLR, MAPK, PI3K, NF-κB, and JAK-STAT signaling pathways. Several studies have demonstrated that the MAPK, PI3K, and NF-κB signaling pathways can increase immunity in *Atractylodes macrocephala* and *Cordyceps sinensis* polysaccharides. The TLR signaling pathway is also a key mechanism by which Cordyceps polysaccharide can improve immune function. The heat map of the effect of tonic Chinese herbal medicine polysaccharides on the immune system is shown in Figure 4.

## 8. Conclusions

The mechanism by which Chinese herbal medicine polysaccharides regulate immune function is becoming clearer with the advancement of science and technology. However, the current research still has many flaws. A range of low-immunity animal models can be used to improve the animal models of tonic Chinese herbal medicine polysaccharides for treating low immunity. The most beneficial tonic Chinese herbal medicine polysaccharides have been reported in the literature to enhance immunity by activating signaling pathways. However, the reported signaling pathways are primarily concentrated in the TLR4 and NF-κB signaling pathways, which can be used to study the effects of other signaling pathways on immune function and clarify the mechanism of increasing immunity. Additionally, there are currently few studies on the adverse reactions of traditional Chinese medicine polysaccharides, but immunotoxicity remains a concern. Therefore, it is essential to conduct more research on its safety. In tonic Chinese herbal medicine, increasing immunity has a variety of theoretical foundations. We can investigate the modern connotation of tonic Chinese herbal medicine and use modern science and technology to examine the in-depth mechanism of tonic Chinese herbal medicine polysaccharides to increase immunity. It establishes a scientific and theoretical foundation for researching, developing, and clinically using novel polysaccharide tonic Chinese herbal medicines.

## Figures and Tables

**Figure 1 molecules-28-07355-f001:**
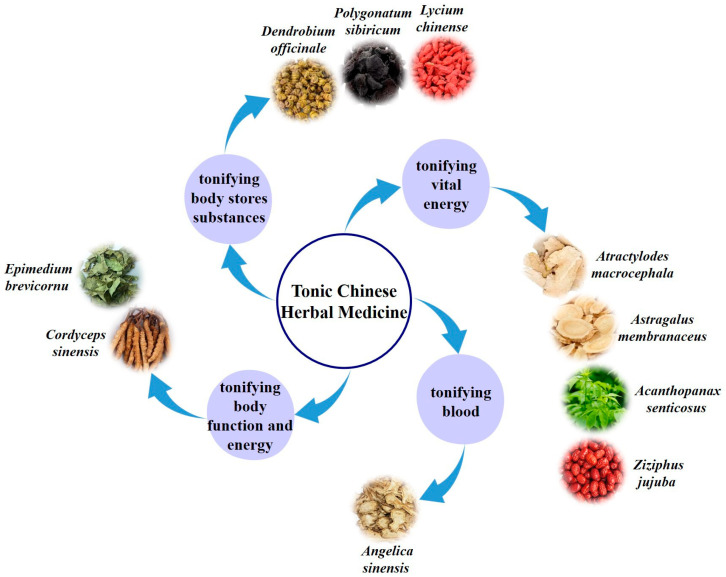
Categories of tonic Chinese herbal medicine. According to the different effects of various traditional Chinese medicine, it can be divided into four categories: tonifying vital energy, tonifying blood, tonifying body function and energy, and tonifying body stores substances.

**Figure 2 molecules-28-07355-f002:**
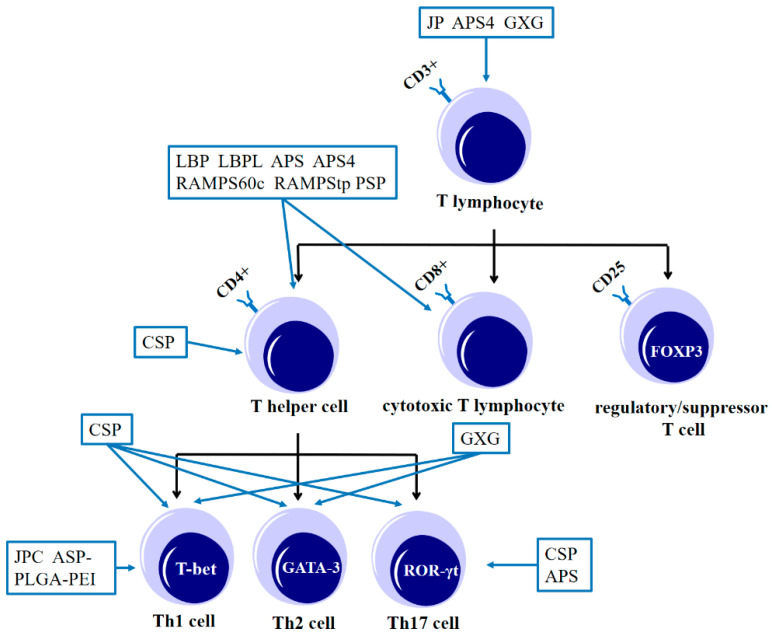
The effect of tonic Chinese herbal medicine polysaccharide on T lymphocytes. JP, JPC: polysaccharides from *Ziziphus jujuba*; APS, APS4: polysaccharides from *Astragalus membranaceus*; CSP: polysaccharides from *Cordyceps sinensis*; GXG: a polysaccharide from *Dendrobium officinale*; LBP, LBPL: polysaccharides from *Lycium chinense*; ASP-PLGA-PEI: polysaccharides from *Acanthopanax senticosus*; RAMPS60c, RAMPStp: polysaccharides from *Atractylodes macrocephala*.

**Figure 3 molecules-28-07355-f003:**
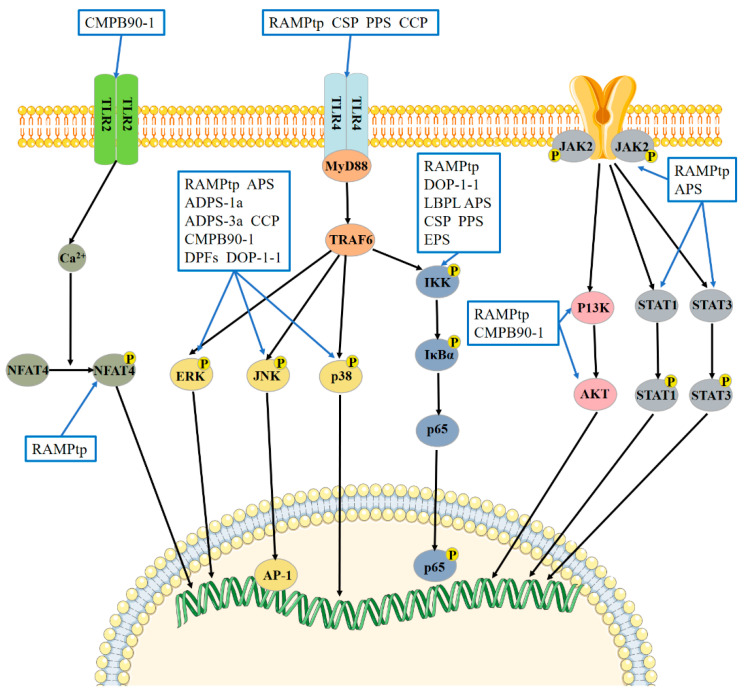
The effect of tonic Chinese herbal medicine polysaccharides on signaling pathways. CMPB90-1, CSP, CCP, and PPS: polysaccharides from *Cordyceps sinensis*; RAMPtp: a polysaccharide from *Atractylodes macrocephala* Koidz.; APS: a polysaccharide from *Astragalus membranaceus*; ADPS-1a, ADPS-3a: polysaccharides from *Angelica sinensis*; DPFs, DOP-1-1: polysaccharides from *Dendrobium officinale*; LBPL: a polysaccharide from *Lycium chinense*; EPS: a polysaccharide from *Epimedium*.

**Figure 4 molecules-28-07355-f004:**
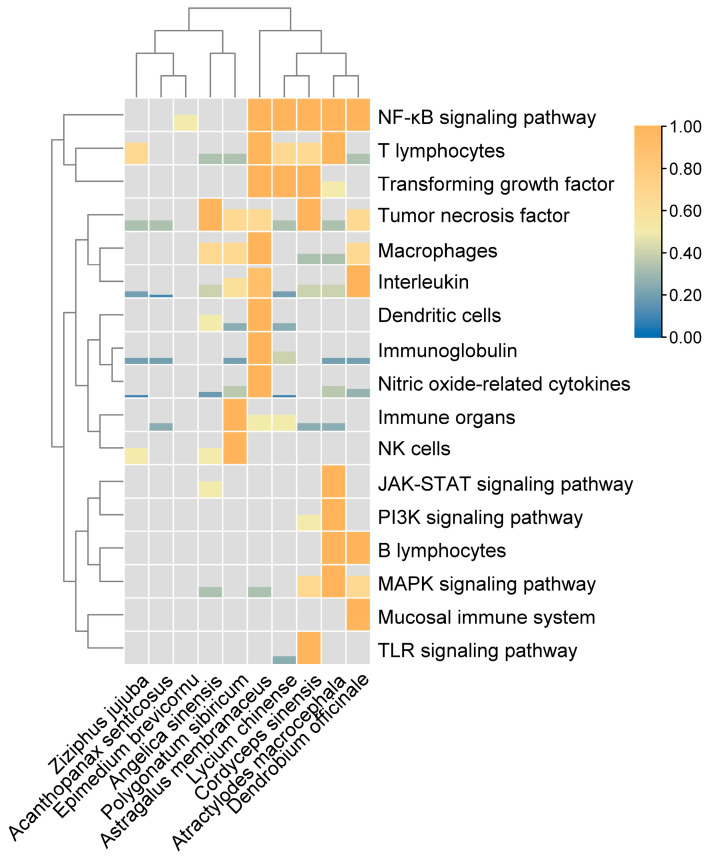
The heat map of the effect of tonic Chinese herbal medicine polysaccharides on immune system. The height of the color region represents the number of researches. The data has been normalized between rows.

**Table 1 molecules-28-07355-t001:** Effects of tonic Chinese herbal medicine polysaccharides on the immune system.

Plant Source	Compound	Administration Mothed	Administration Dosage	Model	Immune Organ	Immune Cell	Cytokine	Ref.
*Atractylodes macrocephala* Koidz.	RAMPtp	N/A	12.5 mg/L	Mouse lymphocytes	N/A	N/A	IL-1α, IL-1β, IL-2, IL-3, IL-4, IL-6, IL-10, IL-12p40, IL-12p70, IL-13, IFN-γ, TNF-α	[47]
	RAMPtp	N/A	25, 50, 100 μg/mL.	Bovine lymphocytes	N/A	N/A	IL-1α, IL-21, IFN-γ, TGF-β1	[74]
	RAMPtp	N/A	25, 50, 100 μg/mL.	Mouse macrophage	N/A	N/A	IL-6, IL-10, TNF-α, iNOS, NO	[32]
	RAMPS	ig	0.25 mL of RAMPS (0.05 g) solution	FMVD O mice	N/A	N/A	IL-6, TNF-α, TGF-β, sIgA	[70]
	RAMPS60c, RAMPStp	ip	0.5 mL 6 mg/mL	ND chicken	thymus, spleen, bursa of Fabricius	CD4+, CD8+	N/A	[20]
*Acanthopanax senticosus*	ASPS	added into forage	1, 2, 4 g/kg	Chicken	N/A	N/A	IgA, IgM	[64]
	ASPS	ig	50, 100, 200 mg/kg	S180, H22, U14 tumor-bearing mice	thymus, spleen	N/A	IL-2, IL-12	[12]
*Ziziphus jujuba*	JP	ig	150, 300, 600 mg/kg	Cyclophosphamide-injected mouse	N/A	CD3+, CD4+, CD8+	IL-2, IL-4, IL-10, IFN-γ, TNF-α, sIgA	[49]
	JPC	ig	100, 200, 400 mg/kg	CSF rat	N/A	CD4+, CD8+, NK cell	IL-2, IL-10	[56]
*Angelica sinensis*	CAP, sCAP2	N/A	500μL 3.125, 1.563, 0.781 μg/mL	Mouse macrophage	N/A	N/A	NO	[88]
	CAP, sCAP2	ip	0.4 mL 0.5, 1, 1.5 mg/mL	Mouse	N/A	N/A	IL-6, IL-10, TNF-α	[88]
	ASP-PLGA-PEI	N/A	31.25 μg/mL	Bone marrow-derived dendritic cell	N/A	N/A	IL-12, TNF-α	[57]
	ISAg	ig	4 mg/mice	B16 melanoma mice	N/A	N/A	IL-12, TNF-α	[31]
	ADPs-1a, ADPs-3a	N/A	ADPs-1a: 100, 200, 400, 800, 1600 μg/mLADPs-3a: 37.5, 75, 150, 300, 600 μg/mL	RAW264.7 cells	N/A	N/A	IL-6, TNF-α, NO	[24]
*Lycium chinense*	LBP	ig	50, 100, 200 mg/kg	Cyclophosphamide-injected mouse	thymus, spleen	N/A	IL-1β, IL-2, IL-6, TNF-α, IFN-γ	[13]
	LBP	ig	0.1 mL/10 g	Mouse	thymus, spleen	N/A	IL-2, IL-6, IFN-γ, TGF-β, IgA, sIgA	[9]
	LBP, LBPF1-5	N/A	1, 3, 10, 30, 100, 300 μg/mL	Mouse lymphocytes	N/A	CD3+, CD19+, CD25	IL-2, IL-4, TNF-α, IFN-γ	[85]
	LBP1-5	ig	250 mg/kg	H22 tumor-bearing mice	N/A	CD4+, CD8+, CD25	TGF-β, IL-10	[94]
*Polygonatum sibiricum*	PSPC, PSPW	ig	200, 400, 800 mg/kg	Spleen deficient mouse	N/A	N/A	IL-2, IL-6, TNF-α, IFN-γ, NO	[25]
	PSP	added into forage	800 mg/kg	Cyclophosphamide-injected chicken	thymus, spleen, bursa of Fabricius	N/A	IL-2, IL-6, IFN-γ, IgG, IgM	[18]
	PSP	ig	100, 200, 400 mg/kg	Cyclophosphamide-injected mouse	thymus, spleen	NK cell, CD4+, CD8+	IL-2, TNF-α	[15]
	PSP, PSP3	ip	PSP: 400 mg/kg PSP3: 100, 200, 400 mg/kg	Cyclophosphamide-injected mouse	thymus, spleen	NK cell	IL-2, IL-4, IL-10, TNF-α	[16]
	PSP	ip	100, 200, or 400 mg/kg	Cyclophosphamide-injected mouse	thymus, spleen	N/A	IL-2, IL-8, IL-10, TNF-α	[17]
*Astragalus membranaceus*	APS	N/A	25, 50, 100, 200 μg/mL	Mouse macrophage	N/A	N/A	IL-1β, IL-6, TNF-α, NO, iNOS	[76]
	APS	N/A	1, 2, 3, 4, 5 mg/mL	Mouse dendritic cell	N/A	N/A	IL-13, IFN-γ	[38]
	APS	ip	0.2 mL 5 μg/mL	ND chicken	spleen, bursa of Fabricius	CD4+, CD8+	IL-2, IL-4, IL-6, IFN-γ	[19]
	APS, APSL	ip	0.5 mL1, 2, 4 mg/mL	OVA mouse	N/A	N/A	IL-6, IFN-γ, IgG, IgG1, IgG2a	[66]
	APS	ip	500 μg	HBV mouse	N/A	CD4+, CTL, DC, Treg	IL-2, IL-4, IFN-γ	[39]
	LMw-APS	ip	100 μg/mice	HSP90C mouse	N/A	N/A	IL-2, IL-4, IL-10, IL-12, IgG1, IgG2b	[67]
	APS	ig	100, 200 and 300 mg/kg	H22 tumor-bearing mice	thymus, spleen	macrophages, NK cell	IL-2, TNF-α, IFN-γ	[26]
	APS	ip	100, 200 mg/kg	4T1 tumor-bearing mice	thymus, spleen	macrophages, lymphocytes, NK cell	IL-2, TNF-α, IFN-γ	[11]
	APS4	ig	150 and 300 mg/kg	S180 tumor-bearing mice	N/A	CD19+ B cell, CD4+, CD8+	N/A	[10]
	APS	ip	10 mg/mL	FUS treated tumor-bearing mice	N/A	N/A	IL-4, IL-10, TNF-α, IFN-γ, IgG1	[38]
	APS	added into forage	0-200 ppm/diet	Necrotizing enteritis chicken	thymus, spleen andbursa of Fabricius	N/A	IL-17	[58]
	APS	added into forage	1 g/kg/diet	SVCV-infected crucian carp	N/A	N/A	IL-1β, IL-8, IL-10, TNF-α, IFN-α, IFN-γ, IgM	[78]
	APS	sc	1.25, 2.5,5 mg/mL	Mouse	N/A	CD4+, CD8+	IL-6, IFN-γ, IgG	[92]
*Dendrobium officinale*	*Dendrobium* CPs	N/A	10, 30, 100, 300, 1000 μg/mL	Mouse macrophage	N/A	N/A	IL-1α, IL-6, IL-10, TNF-α, NO	[23]
	DOP	N/A	50, 150, 300 μg/mL	Mouse macrophage	N/A	N/A	IL-1, IL-6, TNF-α	[80]
	DOP-1-1	N/A	25, 50, 100 μg/mL	THP-1 cell	N/A	N/A	IL-1β, TNF-α	[84]
	DSP	ig	100, 200, 300 mg·kg	Cyclophosphamide-injected mouse	N/A	N/A	IL-6, TNF-α, IFN-γ	[27]
	GXG	ig	50, 200 mg/kg	Mouse	N/A	CD4+, CD8+, B cell, DC cell	IL-1α, IL-1β, IL-2, IL-3, IL-4, IL-5, IL-6, IL-9, IL-10, IL-12, IL-13, IL-17, TNF-α, IFN-γ, sIgA	[50]
	DOP-W3-b	ig	500 mg/kg, 2 g/kg	Mouse	thymus, spleen	N/A	IL-4, IFN-γ	[21]
*Cordyceps sinensis*	CMPB90-1	N/A	15.6, 31.3, 62.5, 125, 250 μg/mL	Mouse lymphocytes	N/A	N/A	IL-2	[30]
	CCP	N/A	2, 20, 100 μg/mL	Mouse macrophage, BMDMs	N/A	N/A	IL-6, TNF-α, NO	[86]
	CSP	ig	25, 50, 100 mg/kg	Cyclophosphamide-injected mouse	N/A	T lymphocytes	IL-17, IL-21, TGF-β3	[52]
	CSP	ig	25, 50, 100 mg/kg	Cyclophosphamide-injected mouse	N/A	N/A	IL-2, IL-4, IL-6, IL-10, IL-12, IL-13, IL-17, IL-21, TNF-α, IFN-γ, TGF-β3	[55]
	PPS	ig	125, 250, 500 mg/kg	Cyclophosphamide-injected mouse	thymus, spleen	Macrophage, CTL, NK cell	IL-2, IL-12, IFN -γ, IgG	[14]

## Data Availability

All date that support the findings of this study are included within the article.

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
