# Peer review of "The Mechanisms of Polysaccharides from Tonic Chinese Herbal Medicine on the Enhancement Immune Function: A Review"

_molecules, 2023, doi:10.3390/molecules28217355_

Round 1
Reviewer 1 Report
1. Please rewrite abstract and conclusion section, as I could not understand the meaning or sentences properly. Kindly make it clear to understand. Eg. Weak syndrome, what does it mean??
2. Table should not be a part of conclusion. Table1 from conclusion may be shifted to section 3 or 4 as per suitability.
3. Future prospects section should come before conclusion, rearrange the text.
Some additional, specific comments such as:
1. Authors addressed the Mechanisms of Polysaccharides from Tonic Chinese Herbal Medicine on Enhancement Immune Function, however, no comments on adverse effects if any were included.
2. The review article has well elaborated studies specifying mechanisms of immune enhancement in different model systems by Polysaccharides from Tonic Chinese Herbal Medicine.
3. The relevant articles pertaining to mechanisms were incorporated but it is again suggested to include a paragraph highlighting the immunotoxicity of Polysaccharides from Tonic Chinese Herbal Medicine.
4. As it is a review article, it is advised to present in systematic review manner following PRISMA guidelines, specifying how the literature was searched, how many studies were selected, and how many studies were excluded with criteria.
5. Conclusion drawn seems to be fine but need clarity in writing some sentences are not clear
eg the mechanisms of action are various can be written as with multiple modes of action.
Eg. stable composition, safe green, and no pollution. Need to rewrite. Like non toxic
6. The references are appropriate.
7. As mentioned Table1 is given in the conclusion section, it's unusual. Conclusion cannot contain tables in my opinion.
The meaning of sentences is not clear specially in abstract and conclusion section, it is advised to take help from english language expert.
Reviewer 2 Report
Comments for molecules-2642927
The manuscript reviews the regulatory effects of many kinds of traditional Chinese medicine polysaccharides on immune organs, immune cells, and immune-related cytokines and explores the immune response mechanism of many kinds of traditional Chinese medicine polysaccharides. The content of the manuscript is interesting, but there are some problem in manuscript should be significantly revised before the manuscript been considered for publication.
Substantial revisions
Q1: Title: Tonic seems redundant and can be deleted.
Q2: Abstract & Keywords: Tonic Chinese herbal medicine seems to be a more suitable alternative to Nourishing Chinese herbal medicine.
Q3: In Fig. 1:
(1) Tonifying seems to be a more suitable alternative to Nourishing.
(2) Was the classification of Types of tonic Chinese herbal medicine based on any previous references ?
Q4: Line 80 : It seems that "immune organs" should be changed to "organs of the immune system" which is more appropriate.
Reviewer 3 Report
1. Rephrase line no 17to 19.
2. Re write the key words avoid the use of words in the title/ abstract in keywords
3. Rewrite line no 51-54
4. Does the tonic Chinese polysaccharide, induce hyper immune system, because in point of Corona virus, the immunity causes the death in many cases through the event of cytokine storm.
5. Elaborate the weak syndrome
6. Rewrite the introduction scientifically, it doesn’t sound so. The content discussed was too vague.
7. The images in the figure 1 isn’t clear change the pixel and re draw them as per the journal guidelines
8. The terms used in the introduction isn’t globally understandable example: qi, yang…
9. Rewrite the figure 1 legend.
10. Rephrase line no 81, cytokines do not originate from immune organs kindly write in scientific terms
11. Does the MS have any focus on the effect of TC herbal medicine in birds
12. Rewrite line no 122
13. In line 241 “Cytokines are small molecules that transmit information between immune cells.” Verify the term.
14. Cytokines do not transmit information. Kindly verify the statement and rewrite the statement
Extensive English language editing required
Round 2
Reviewer 1 Report
Authors responded positively to the comments/suggestions, thanks. The authors revised the manuscript substantially as per suggestions.
Would like to request one more change i.e. conclusion should be only one paragraph (line 499 to line 505). The rest part may be given as separate section such as Mechanistic aspects of tonic chinese herbalmedicine polysacchrides on immune system (Line 506 toline 556, including fig 4).
Rest is fine with me.
Thanks.
Author Response
Thanks for your suggestion. We have divided the conclusion section into discussion and conclusion to meet the manuscript requirements. We change to our manuscript within the document were highlighted by using yellow colored text.
Reviewer 3 Report
Large number of readers wont be able to understand Chinese language. To overcome this some words specifically described in native language can be avoided.
Author Response
We have replaced the Chinese technical terms in the manuscript with commonly used words to improve the readability of the manuscript. We change to our manuscript within the document were highlighted by using yellow colored text.